# Synthesis and Characterisation of Novel Bis(diphenylphosphane oxide)methanidoytterbium(III) Complexes

**DOI:** 10.3390/molecules27227704

**Published:** 2022-11-09

**Authors:** Shalini Rangarajan, Owen A. Beaumont, Maravanji S. Balakrishna, Glen B. Deacon, Victoria L. Blair

**Affiliations:** 1IITB-Monash Academy, Indian Institute of Technology Bombay, Mumbai 400076, India; 2School of Chemistry, Monash University, Melbourne 3800, Australia

**Keywords:** ytterbium, phosphine oxide, methanide, C-F activation, fluoride cage

## Abstract

Reaction of [YbCp_2_(dme)] (Cp = cyclopentadienyl, dme = 1,2 dimethoxyethane) with bis(diphenylphosphano)methane dioxide (H_2_dppmO_2_) leads to deprotonation of the ligand H_2_dppmO_2_ and oxidation of ytterbium, forming an extremely air-sensitive product, [Yb^III^(HdppmO_2_)_3_] (**1**), a six-coordinate complex with three chelating (OPCHPO) HdppmO_2_ ligands. Complex **1** was also obtained by a redox transmetallation/protolysis synthesis from metallic ytterbium, Hg(C_6_F_5_)_2,_ and H_2_dppmO_2_. In a further preparation, the reaction of [Yb(C_6_F_5_)_2_] with H_2_dppmO_2_, not only yielded compound **1**, but also gave a remarkable tetranuclear cage, [Yb_4_(µ-HdppmO_2_)_6_(µ-F)_6_] (**2**) containing two [Yb(µ-F)]_2_ rhombic units linked by two fluoride ligands and the tetranuclear unit is encapsulated by six bridging HdppmO_2_ donors. The fluoride ligands of the cage result from C-F activation of pentafluorobenzene and concomitant formation of *p*-H_2_C_6_F_4_ and *m*-H_2_C_6_F_4_, the last being an unexpected product.

## 1. Introduction

Phosphane oxide ligands, and especially triphenylphosphane oxide, are popular ligands for rare earth metals [1,2,3,4]. Complexation is aided by the oxophilic nature of lanthanoid ions and is enhanced by the polarity of the P^+^–O^−^ bonds. The spear-like donor functionality limits steric repulsion close to the metal but provides steric protection further away. Some use has been made of these ligands in solvent extraction/separation of lanthanoids [5]. They have also been used in the stabilisation and crystallization of highly reactive organolanthanoids, aryloxides and organoamides, including pyrazolates [6,7,8,9,10,11,12,13]. Interestingly, [YbCp_2_(dme)] (Cp = cyclopentadienyl, dme = 1,2 dimethoxyethane) undergoes complexation with Ph_3_PO without the occurrence of a redox reaction to give [YbCp_2_(OPPh_3_)_2_] [6,8]. More recently attention has turned to chelating phosphane oxides with bis(diphenylphosphano)methane dioxide attracting particular attention [14,15,16,17,18,19,20]. Still more recently, reactions of yttrium and lanthanum tris(bis(trimethylsilyl)amide) complexes with bis(diphenylphosphano)- and bis(dimethylphosphano)-methane dioxide led to the deprotonation and formation of homoleptic rare earth complexes of bis(phosphane-oxide)methanide species (Figure 1, previous work, reaction 1a) [14]. We now report that the reaction of dicyclopentadienylytterbium(II) with bis(diphenylphospano)methane dioxide (H_2_dppmO_2_) does not lead to a simple complexation giving [YbCp_2_(H_2_dppmO_2_)], as with Ph_3_PO [6], but rather both deprotonation and oxidation occur, to give the tris{bis(diphenylphosphane oxide)methanido}ytterbium(III) complex, [Yb(HdppmO_2_)_3_] (**1**) (Figure 1, present work, reaction 1b). The compound **1** was also prepared by a redox transmetallation (RTP) reaction between Yb metal, Hg(C_6_F_5_)_2_, and bis(diphenylphosphano)methane dioxide (Figure 1, present work, reaction 1b). However, the reaction of bis(pentafluorophenyl)ytterbium and bis(phosphano)methane dioxide not only yields **1**, but also forms a (hexafluorido)tetraytterbium(III) cage [Yb_4_(µ-HdppmO_2_)_6_(µ-F)_6_] (**2**) by C-F activation of pentafluorobenzene (Figure 1, present work, reaction 1c). The paper provides new synthetic paths to bis(phosphine oxide)methanide complexes and a remarkable fluoride-bridged cage.

## 2. Results and Discussion

### 2.1. Synthesis and Characterisation of [Yb(HdppmO_2_)_3_] ***1***

Initially, an attempt was made to prepare [YbCp_2_(H_2_dppmO_2_)] by reaction of [YbCp_2_(dme)] (dme = 1,2-dimethoxyethane) with an equimolar amount of bis(diphenylphosphano)methane dioxide (H_2_dppmO_2_) (Figure 2, Route 1), analogous to the reaction conditions used in the preparation of [YbCp_2_(OPPh_3_)_2_] [6]; instead, an oxidation reaction occurred and the tris{bis(diphenylphosphane oxide)methanido}ytterbium(III) complex, [Yb(HdppmO_2_)_3_] **1** was isolated, together with CpH and unreacted [YbCp_2_(dme)]. The ^31^P{^1^H} NMR spectrum revealed a considerably deshielded ^31^P{^1^H} signal at 39.1 ppm (*c.f* H_2_dppmO_2_ 24.5 ppm) (See Appendix A) and the ^1^H NMR spectrum showed the formation of CpH indicative of deprotonation of H_2_dppmO_2_. In addition, the presence of unreacted [Yb Cp_2_ (dme)] was also detected in the reaction mixture. (See Appendix A).

To facilitate a rational synthesis and effective utilization of [YbCp_2_(dme)], the reaction stoichiometry was increased to a 3:1 ratio of H_2_dppmO_2_: [YbCp_2_ (dme)], resulting in the isolation of crystalline **1** in 65% yield. NMR spectroscopy of the reaction mixture shows complete consumption of the [YbCp_2_(dme)] (See Appendix A). Thus, both protolysis and redox processes are observed in the reaction of [YbCp_2_(dme)] with H_2_dppmO_2_ (Figure 2, route 2). With the same mole ratio the reaction was carried out in an NMR tube in d_8_-toluene, revealing a resonance at 4.79 ppm attributable to H_2_ gas formation [21,22,23] (See Appendix A). The composition of **1** was established by microanalysis and high-resolution mass spectrometry which revealed the [M+H]^+^ ion (see Experimental Section 3.4).

The infrared spectrum of the extremely air- and moisture-sensitive **1** shows very strong ν(P=O, cm^−1^) absorptions at 1154, 1120, 1081 and 1061 (Figure 1, top), which are very different from those of free ligand (1212, 1197 and 1176 cm^−1^) (Figure 1, bottom). A strong band of the ligand at 1111 cm^−1^ can be assigned to the X-sensitive mode q, involving P-C stretching [24,25]. This is not unduly shifted or enhanced in intensity on coordination as indicated by the spectra of Ph_3_P=O complexes [24] and presumably contributes to the intensity of the 1120 cm^−1^ band of the complex. Weaker C-H in-plane deformation modes of the ligand at 1083 and 1027 cm^−1^ are unlikely to be affected by coordination. The shifts of ν(P=O) are reminiscent of the shift of ν(C=O) of the diketo form of beta-diketones to ν_as_(C=O) of β-diketonate complexes [26]. The absorption bands of **1** (see Experimental Section 3.4) are somewhat similar to those listed for [M(HdppmO_2_)_3_] [14] (M = Y, La), where absorptions at 1148 (M = Y) and 1076, 1050 cm^−1^ (M = La) are assigned to ν(P=O). On the other hand, in the complex [Ln(H_2_dppO_2_)_4_]3^+^, containing neutral H_2_dppmO_2_, the absorptions at 1197 and 1098 cm^−1^ are assigned to ν(P=O) [15], though the latter may be a slightly shifted q vibration, thereby indicating that the absorptions are little affected when the ligand is neutral. Similarly, [Ga(*t*Bu)_3_(H_2_dppmO_2_)] has strong bands at 1210, 1187, 1170 (ν P=O) and 1110 (q), and medium C-H deformation bands at 1075 and 1028 cm^−1^, all near free ligand values, but the complex of the deprotonated ligand [Ga(*t*Bu)_2_(HdppmO_2_)] has strong bands at 1195, 1165, 1110, 1076, 1060 cm^−1^ [27], which are similar to those of **1**, apart from the first value.

### 2.2. Solid State Structure of [Yb(HdppmO_2_)_3_] ***1***

Slow evaporation of a solution of **1** in acetonitrile yielded polychromatic blocks of compound **1**·MeCN which crystallised in the triclinic space-group (P ī) with a molecule of lattice acetonitrile in the asymmetric unit. A single-crystal X-ray structure determination of **1** shows a six-coordinate complex, having a slightly distorted octahedral stereochemistry (Figure 2) for ytterbium with cis and trans O–Yb–O angles in the ranges of 85.01(6)–93.23(6)° and 174.1(6)–176.86(6)°, respectively (See Appendix A). Although the structure is similar to those of [M(HdppmO)_3_] (M = La or Y) [14], the structures are not isomorphous, possibly owing to the presence of lattice solvent in **1**·MeCN. The Yb-O bond lengths (2.1953(16)–2.2263(14)) Å are in the range of comparable reported amide complexes, namely, (a) [Yb{N(PO(OC_6_H_5_)_2_)_2_}_3_] [28]; (2.199(6)–2.219(6) Å) and (b) [Yb{N(PO(C_6_F_5_)_2_)_2_}_3_] [29]; (2.18(2)–2.28(2) Å). The Yb-O bond lengths of **1** are shorter than those in [Yb(H_2_dppmO_2_)_3_Cl]^+^ where the ytterbium atom is bound to neutral H_2_dppmO_2_ (Yb-O 2.250(3)–2.338(3) Å) [16]. The P-O and P-C bond lengths in **1** range between 1.5152(16)–1.5317(16) Å and 1.706(2)–1.714(2) Å, respectively, which are different from those of the free ligand H_2_dppmO_2_ [30] (P-O: 1.486(2) Å and P-C 1.815(2) Å) indicating partial double bond character for P-O and P-C bonds in the chelate ring as reported in the case of the La and Y analogues [(M(HdppmO_2_)_3_)] [14]. However, the P-O double bond character is reduced on coordination, whilst the double bond character of the P-C bond is considerably increased, owing to charge delocalization on deprotonation.

### 2.3. Properties of [Yb(HdppmO_2_)_3_] ***1*** in Solution

The ^1^H NMR spectrum of compound **1** in CD_2_Cl_2_ (see Appendix A) is broad owing to the paramagnetic nature of the complex. The two broad singlets observed at −3.26 and 7.45 ppm are assigned to methanide and aromatic protons, respectively. The methanide CH^−^ proton is upfield shifted by ~6 ppm relative to the CH_2_ resonance of the free ligand (3.56 ppm) [31] indicating it is highly shielded in comparison to other [M(HdppmO_2_)_3_] complexes (M = Y (δH = 2.26 ppm), M = La (δH = 2.28 ppm)) [14] presumably owing to Yb^III^ paramagnetism. This pronounced shielding of the methanide proton was supported by Natural Bond Orbital (NBO) analysis on complex 1 (see Appendix A for full details). It was observed that Natural Population Analysis (NPA) revealed substantial negative natural charge of −1.43 at the methanide carbon (see Appendix A), where *q*C of CH^−^ was significantly more negative than the aromatic carbon atoms attached to the phosphorus atom (−0.395 to −0.425) in **1**, implying high shielding of the CH^−^ proton in the ^1^H NMR spectrum, as observed experimentally (See Appendix A). The positive natural charge for Yb(III) was calculated to be 1.84 which is in the range as mentioned in the literature [32]. In the series (Ln(HdppmO_2_)_3_) Ln = La, Y [14], Yb (*this work*) the calculated charge decreases 2.30, 2.00, 1.84, respectively, as the size of the ion decreases, reflecting increased ligand to metal charge transfer in the same sequence. The ^31^P{^1^H} NMR spectrum of complex **1** shows a singlet at 39.1 ppm with a coordination shift of 14.2 ppm (H_2_dppmO_2_; 24.9 ppm), and is similar to the reported [M(HdppmO)_3_] (M = Y (37.8 ppm), La (35.4 ppm)) [14], hence the effect of deprotonation outweighs paramagnetic effects in ^31^P{^1^H} NMR spectroscopy (see Appendix A).

To examine the solution behavior further, ^1^H NMR spectra were recorded for complex **1** in CD_2_Cl_2_ from 25 °C to −75 °C (Figure 3). On lowering the temperature from 25 °C to 5 °C, coalescence was observed until −15 °C when the peaks were resolved (Figure 3). At −55 °C, sharp peaks are seen with no further changes below. This phenomenon can be rationalised by a conformational equilibrium [33] in **1**. At 25 °C the conformers are found to be highly fluxional, while at −55 °C the dynamic equilibrium between the conformers ceases, whereas in case of the reported [M(HdppmO_2_)_3_] (M = Y, La) complexes, they are found to be fluxional even at −80 °C [14].

The presence of conformers is further supported by close analysis of the solid-state structure. Three six-membered YbO_2_P_2_C metallacycles, present in **1**, exhibit a combination of boat and twist-boat conformers within the solid-state (Figure 4 and Figure 5). The torsion angles of the O-P-P-O atoms in the metallacycle are all significantly different for all the three metallacycle rings in both the boat and twist-boat conformations (Table 1).

This indicates that at 25 °C there is a rapid intramolecular dynamic process within complex **1**. As the temperature reaches −15 °C, the spectrum indicates the interconversion process slowing down between the possible conformers. At −55 °C the intramolecular dynamic process ceases as resolved peaks are distinctly observed. This indicates the possibility of phenyl groups frozen in the axial and equatorial positions at −55 °C, where there are mirror image peaks (See Appendix A). Thereby, we can conclude that at 25 °C, complex **1** is highly fluxional, but at −55 °C the fluxionality is lost. Variable temperature ^31^P{^1^H} NMR spectroscopy did not show much variation within the temperature range of 25 °C to −75 °C other than a slight shift to a higher frequency (See Appendix A).

### 2.4. Synthesis of ***1*** by the RTP Method

In order to examine the redox transmetallation protolysis (RTP) protocol [34] an excess of metallic Yb was treated with Hg(C_6_F_5_)_2_ and H_2_dppmO_2_, in THF at room temperature for 15 min (Figure 3). Analysis of the reaction mixture by ^19^F{^1^H} NMR spectroscopy revealed peaks at −139.8, −155.5 and −163.4 ppm corresponding to the formation of C_6_F_5_H [35] and a small peak at −140.2 ppm, assigned to *p*-H_2_C_6_F_4_ [35] (See Appendix A). The ^31^P{^1^H} NMR spectrum showed a single resonance at 39.8 ppm indicative of complex **1**. Recrystallisation of this extremely air- and-moisture sensitive compound from dry acetonitrile gave polychromatic blocks of complex **1** in a 71% yield. This simple one-pot procedure should be applicable to all rare earth elements, and does not require a prior synthesis of metal precursors, e.g., of tris{bis(trimethylsilyl)amido}-lanthanoid(III) complexes [14]. This is the first synthesis of a methanido-lanthanoid complex by this method, but should be widely applicable, for example to β-diketonates.

### 2.5. Synthesis of [Yb_4_(µ-HdppmO_2_)_6_(µ-F)_6_] ***2***

RTP reactions between Yb metal, Hg(C_6_F_5_)_2_, and a protic reagent LH (LH = a weak protic acid e.g., phenols, amines, CpH, pyrazoles, amidines etc.) proceed through a [Yb(C_6_F_5_)_2_] intermediate, which is protolysed to give pentafluorobenzene and Yb^II^L_2_, but when Yb^III^L_3_ is the final product, YbL_2_ is further oxidised by Hg(C_6_F_5_)_2_ to give YbL_2_(C_6_F_5_), which is protolysed to give YbL_3_ [34]. Since [Yb(C_6_F_5_)_2_] is a key intermediate in RTP syntheses, we examined the reaction of independently prepared [Yb(C_6_F_5_)_2_] [36,37] with H_2_dppmO_2_. It also provides a comparison with the reactions of [YbCp_2_(dme)] (as discussed in Section 2.1).

[Yb(C_6_F_5_)_2_] was prepared by reacting an excess of Yb with Hg(C_6_F_5_)_2_ (Figure 4). Addition of the filtered [Yb(C_6_F_5_)_2_] solution to H_2_dppmO_2_ in THF in a 1:3 ratio afforded a solution colour change from red/orange to colorless over 20 min. The ^19^F{^1^H} NMR spectrum of an aliquot of the reaction mixture revealed seven peaks, identified as corresponding to the major products; pentafluorobenzene (−139.6, −155.59 and −163.5 ppm) [35] and 1,2,4,5-tetrafluorobenzene (−140.7 ppm) [35] and minor product, 1,2,3,5-tetrafluorobenzene (−114.2, −133.1 and −167.5 ppm) [38] (See Appendix A). The corresponding ^31^P{^1^H} NMR spectrum of the reaction mixture shows a sharp peak at 39.1 ppm corresponding to compound **1**. (see Appendix A).

A small amount of colourless blocks was deposited from a concentrated reaction mixture aliquot added to *d_6_*-benzene. The crystals were subjected to X-ray single crystal structure determination which revealed the presence of the C-F activation product [Yb_4_(µ-HdppmO_2_)_6_(µ-F)_6_]**·**6THF**·**3C_6_D_6_ **2** (Figure 5). Compound **2** is obtained in low yield and is insoluble in most deuterated organic solvents (*d_6_*-benzene, *d_8_*-thf, CD_3_CN, CD_2_Cl_2_ and *d_8_*-toluene) inhibiting analysis in the solution state.

### 2.6. Solid State Structure of [Yb_4_(µ-HdppmO_2_)_6_(µ-F)_6_] ***2***

The X-ray structure of **2** (Figure 6) shows four ytterbium centres bridged by six HdppmO_2_ and six fluoride ions with each metal centre adopting slightly distorted octahedral geometry. The tetra-nuclear structure of **2** can be regarded as a dimer of dimers. It is based on a Yb_4_F_6_ cage with alternating single and double bridges between adjacent ytterbium atoms. Six bridging (O,O’) HdppmO_2_ ligands form the periphery of the cage and also alternate between double and single ligand bridges with the pairs corresponding to single fluoride bridges and *vice versa.* Two [Yb(µ-F)]_2_ rhombic units are linked by two fluoride ligands. By contrast, the iso-stoichiometric cages reported earlier, [Yb_4_F_6_L_6_] (L = *p*-HC_6_F_4_N(CH_2_)_2_NR_2_; R = Me or Et), consist of one rhombic [Yb(µ-F)]_2_ unit linked on both sides by two metal centres via four M—F—M units, and the amide ligands are chelating not bridging [39]. In the case of **2**, the Yb-O bond lengths (2.162(2)–2.189(2)Å) are slightly shorter than those of compound **1** (2.1953(16)–2.2263(15) Å), probably due to the electronegative fluoride co-ligands. The P-O bond lengths (1.515(2)–1.524(2) Å) are relatively longer and P-C_H_ lengths (1.695(4)–1.712(4) Å) are relatively shorter than those in the free ligand H_2_dppmO_2_ (P-O 1.486(2)Å and P-C_H_ 1.812(2) Å) [30] indicating partial double bond character of P-O and P-C bonds. The P-O and P-C bond lengths are very close to those of **1**. The Yb-F bond lengths in [Yb(µ-F)]_2_ are slightly shorter than those which link two rhombic units (Figure 5). Overall, the Yb-F bond lengths in compound **2** are in the range of previously reported complexes (*c.f.* [Yb_4_ (*p-*HC_6_F_4_N(CH_2_)_2_NMe_2_)_6_F_6_], 2.150(2)–2.190(2) Å) [39].

Inhibited by low solubility, **2** was also characterized by elemental analysis (see Experimental Section 3.4). The formation of **2** by C-F activation, utilising an organolanthanoid, is very rare in being accompanied by the formation of 1,2,3,5-tetrafluorobenzene along with 1,2,4,5-tetrafluorobenzene. Ytterbium (and Eu) can activate C_6_F_5_H with the formation of the observed *p*-C_6_F_4_H_2_, though the reaction is slow [40,41]. However, in the present reaction, the Yb metal is separated before C_6_F_5_H is generated by protolysis. In previous work, 1,2,3,4-tetrafluorobenzene formation has been observed in C-F activation reactions that accompany the redox transmetallation between Sm metal and Hg(C_6_F_5_)_2_ [42], but there was no report of the formation of *m*-C_6_F_4_H_2_. A possibility source of two tetrafluorobenzenes is that [Yb(HdppmO_2_)_2_], the initial product of protolysis of [Yb(C_6_F_5_)_2_], reacts unselectively with the *m*-F and *p*-F atoms of C_6_F_5_H by single electron transfer to generate [Yb(HdppmO_2_)_2_F] and isomeric *m*- and *p*-C_6_H_1_F_4_^•^ radicals which capture hydrogen radicals from the solvent to give *m*- and *p*-H_2_C_6_F_4_. Complex **2** is then formed by the rearrangement of the [Yb(HdppmO_2_)_2_F] (Figure 5).

## 3. Materials and Methods

### 3.1. General Procedures

All the lanthanoid metals and lanthanoid(II) and (III) products are highly air- and moisture-sensitive, hence operations were carried out under nitrogen using standard Schlenk-line and glovebox techniques. Ytterbium metal was purchased as metal ingots from Santoku or Eutectix. [YbCp_2_(dme)] [43], H_2_dppmO_2_ [44], Hg(C_6_F_5_)_2_ [45] and [Yb(C_6_F_5_)_2_] [36,37] were synthesised by literature procedures. THF was dried and deoxygenated by refluxing over Na metal and distillation from sodium benzophenone ketyl, whereas MeCN was dried and deoxygenated by refluxing over and distillation from calcium hydride. The dried solvents were stored over 4 Å molecular sieves. Infrared spectra (4000–650) cm^−1^ were obtained with Nujol mulls between NaCl plates with a Perkin-Elmer 1600 FT-IR spectrometer. Room temperature (25 °C) ^1^H and ^31^P{^1^H} NMR spectra were recorded with a Bruker DPX 300 instrument using d_8_-toluene, C_6_D_6_ or CD_2_Cl_2_. The solvents were dried over 4 Å molecular sieves for 48 h, and resonances were referenced to residual hydrogen-atom resonances of the deuterated solvent.

### 3.2. Single Crystal X-ray Structure Determination

Crystals for X-ray structure analysis were grown using saturated solutions in acetonitrile (**1**), or THF-C_6_D_6_ (**2**). Crystals **1** and **2** were immersed in paratone, and were measured on a Rigaku SynergyS diffractometer. The SynergyS operated using microsource Cu-*K*α radiation (λ = 1.54184 Å) at 123 K. Data processing was conducted using the CrysAlisPro.55 software suite [46]. Structural solutions were obtained by ShelXT [47] and refined using full-matrix least-squares methods against F^2^ using SHELXL [48], in conjunction with Olex2 [49] graphical user interface. All hydrogen atoms were placed in calculated positions using the riding model.

### 3.3. Computational Studies

All calculations reported were performed using the Gaussian 09 suite of programs. The coordinates for the calculations were directly taken from X-ray structure of compound **1**. The calculations were performed on complex **1** using the basis set def2-TZVP [50,51,52,53,54]/6-31G (d, p) [50,51,52,55,56] for carrying out natural bond analysis and MO6L was chosen as functional for our study. def2-TZVP is a basis set used for Yb metal [57] and the 6-31G (d, p) basis set is used for C, H, O and P. Full details are in the Appendix A.

### 3.4. Experimental Section

#### 3.4.1. Preparation of [Yb{(Ph_2_PO)_2_CH}_3_] (**1**)

Method 1: [YbCp_2_(dme)] (30 mg, 0.076 mmol) and H_2_dppmO2 (95.2 mg, 0.228 mmol) in dry THF (5 mL) at room temperature, were stirred for 30 min. The solution was evaporated and 10 mL of dry MeCN were added. The volume of the solution was concentrated by half to obtain extremely air- and moisture-sensitive polychromatic crystals of **1**, 70 mg, 65% yield. M.P decomp. temp at 380 °C. ^1^H NMR (at 25 °C 400 MHz, CD_2_Cl_2_) δ 7.22 (s, 60H), −3.14 (s, 3H). ^1^H NMR (at −75 °C 400 MHz, CD_2_Cl_2_) δ 13.71–12.68 (m, 11H), 8.75 (s, 13H), 8.18 (s, 6H), 5.91 (s, 6H), 5.28 (s, 13H), 3.18 (s, 11H), −3.98 (s, 3H). ^31^P{^1^H} NMR (at 25 °C, 162 MHz, CD_2_Cl_2_) δ 39.20. HRMS(*m*/*z*) calcd. for C_75_H_63_O_6_P_6_Yb 1420.2526 (M+1); found 1420.2518 (M+1, 0.08%). IR spectrum (1700–650) cm^−1^/Nujol 1676(w), 1590(w), 1573(w), 1438(s), 1308(m), 1262(s), 1152(s), 1120(vs), 1081 and 1062(vs), 1026(s), 999(w), 947 and 932(vs), 812(s), 745(s), 722(s), 692(s). Elemental analysis calcd (%) for C_75_H_63_O_6_P_6_Yb (loss of MeCN lattice solvent): C 63.47, H 4.47; found C 63.50, H 4.65.

Method 2: Yb metal (166.2 mg, 0.960 mmol), Hg(C_6_F_5_)_2_ (128.3 mg, 0.239 mmol) and H_2_dppmO_2_ (200 mg, 0.480 mmol), stirred for 15 h at room temperature in dry THF. The reaction mixture was filtered and dried under vacuum. Anhydrous MeCN was added to the crude product followed by concentration of the solution to yield extremely air- and moisture-sensitive polychromatic blocks of compound **1**. The unit cell values of the crystals (by SCXRD studies) matched that of compound **1** (Yield = 241 mg, 71%).

Full crystal and refinement data for complexes **1** and **2** can be found in the Appendix A.

#### 3.4.2. Preparation of [Yb_4_{(Ph_2_PO)_2_CH}_6_F_6_] (**2**)

Yb metal (166.2 mg, 0.960 mmol) and Hg(C_6_F_5_)2 (128.3 mg, 0.239 mmol) in dry THF were stirred for 2 h at room temperature and the reaction mixture was filtered into a THF solution of H_2_dppmO_2_ (300 mg, 0.717 mmol) and stirred for 20 min at room temperature. The reaction mixture was filtered, and an aliquot of the solution in C_6_D_6_ was subjected to NMR analysis in a Young’s NMR tube. The NMR solution yielded extremely air- and moisture-sensitive colourless block crystals identified as compound **2**. Yield of **2** (17 mg, 2.2%). An attempt was made to obtain the infrared spectrum of **2** but it has limited stability in Nujol and turns black gradually. Gradual decomposition is also apparent in the X-ray oil. ^19^F{^1^H} NMR of the reaction mixture (376 MHz, C_6_D_6_) δ −139.6, −155.59 and −163.5 (C_6_F_5_H), −140.7 (1,2,4,5-C_6_F_4_H), and −114.2, −133.1 and −167.5 (1,2,3,5-C_6_F_4_H). Elemental analysis calculated (%) for C_150_H_126_O_12_P_12_Yb_4_F_6_ (loss of lattice solvent 3 × C_6_D_6_ + 6 × thf) C 54.62, H 3.85 found C 54.59 H 3.89. HRMS (*m*/*z*) calculated for Yb(HdppmO_2_)_3_ + Ph_2_P(O)OH + H: 1638.3031 found 1638.3168. The rest of the reaction solution was evaporated to dryness and anhydrous MeCN was added to the crude product followed by concentration of the solution to yield extremely air- and moisture-sensitive colourless block crystals of compound **1** (Yield = 230 mg, 68%).

## 4. Conclusions

The complex, [Yb(HdppmO_2_)_3_] **1**, with a chelating O,O’-donor set of bis(diphenylphosphane oxide)methanide ligands, has been obtained by oxidative protolysis of [YbCp_2_(dme)] with H_2_dppmO_2_, by a redox transmetallation/protolysis reaction between Yb metal, Hg(C_6_F_5_)_2_ and H_2_dppmO_2_, and by oxidative protolysis of [Yb(C_6_F_5_)_2_] with H_2_dppmO_2_. The latter reaction also yielded a novel fluoride-bridged cage, [Yb_4_(µ-HdppmO_2_)_6_(µ-F)_6_] **2**, with a core of Yb atoms alternately singly and doubly bridged by fluoride, YbFYbF_2_YbFYbF_2_. The fluoride ligands are derived from C-F activation of pentafluorobenzene giving *p*-H_2_C_6_F_4_ and, unexpectedly, *m*-H_2_C_6_F_4_. The RTP synthesis provides a simple one pot route for all rare earth tris{bis(diphenylphosphane oxide)methanides}, and also for other analogous phosphine oxide-derived species and potentially methanidolanthanoid species in general.

## Data Availability

Crystal data can be obtained free of charge from The Cambridge Crystallographic Data Centre: CCDC 2209922 for **1** and 2209923 for **2**.

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
