# Peer review of "Synthesis and Characterisation of Novel Bis(diphenylphosphane oxide)methanidoytterbium(III) Complexes"

_molecules, 2022, doi:10.3390/molecules27227704_

Round 1

Reviewer 1 Report

In the manuscript “Synthesis and characterisation of novel bis(diphenylphosphaneoxide)methanidoytterbium(III) complexes” the authors describe the syntheses, structural, and spectroscopic properties of four new compounds. The crystal structure of both compounds is described and determined by single-crystal X-ray diffraction. The one pot synthesis based on redox transmetallation/protolysis is also examined with Ytterbium and bis(diphenylphosphano)methane dioxide (H2dppmO2). The scientific contents are finely presented and gave the contribution to chemistry. Nevertheless, there are a couple of problems that need correction to improve the quality of the manuscript. I recommend the publication of this manuscript after the necessary corrections.

1. Introduction, page 1, lines 28-29: “Interestingly, [YbCp2(dme)] undergoes complexation with Ph3PO without the occurrence of redox [6, 8].” This sentence looks unfinished. It needs to be rephrased.

2    2. In the Introduction of the manuscript is given the formula [YbCp2(dme)] and dme is defined in the abstract, but what is Cp? Since the abbreviation Cp is mentioned in the manuscript several times it needs to be defined at least in the Introduction.

3    3. The importance of this research and results should be emphasized in the Introduction.

4. About Scheme 1 is not clear if the upper part of it is a previous work of the authors of this manuscript or it is a work of others. Besides in the text bellow the citation of the scheme 1 is not clear if it is part of the also cited literature or not. It should be marked like 1a and 1b or similarly. Then the authors can add if it is presented in a previous publication.

5.       Page 3, line 70 [M+H]+ should be corrected to [M+H]+ with + in the superscript.

6.       The Figures S3 and S4 should be presented in the manuscript in part 2.1 below the regarding part of the discussion as one figure (with overlapped spectra).

7.       Page 3, line 94 the formula 1•MeCN should be corrected and written with a middle dot instead of a bullet: 1∙MeCN.

8.       Figures 1 and 5: The selected bond lengths and angles should be given in tables and nowise in the figure capture. On Figure 1 oxygen atoms are purple and phosphorus atoms are red, but on the other figures is opposite. This one should be changed and all the figures with structures need the same color code.

9.       Page 7, line 198: What is “a protic reagent LH”? The redox transmetallation/protolysis (RTP) is shortly defined under the previous subtitle, but the protic reagent is new, and it should be also shortly defined (in one sentence like the description of RTP procedure).

10.   The crystal data should be given in the manuscript in the form of a table or completely deleted from the main text and given only in the supplementary information, but it should not be given as a plane text in the experimental part and in the same time as a table in the supplement.

11.   Conclusion: “The complex 1, with chelating (O2) bis(diphenylphosphane oxide)methanide ligands…” – What is here the O2? If the authors would present the donor set of the ligand, than they would say: with chelating OO donor set of bis…., or with chelating 2O donor set…

Reviewer 2 Report

This manuscript describes two new and interesting synthetic pathways to obtain a lanthanide phosphane oxide complex, an increasingly studied family of lanthanide complexes. In this study, the trivalent bis(diphenylphosphane oxide)methanide ytterbium complex was obtained starting from either the divalent Cp2Yb(dme) complex or from Yb metal via redox-transmetallation protolysis. In the latter case, an additional C-F bond activation product was formed and a mechanistic rationale was proposed. All the complexes were nicely characterized by XRD and elemental analysis and for the major product by variable temperature NMR, ir and HRMS. The observation of conformers was supported by theoretical calculations. 

Please find below some comments the authors may take into consideration when preparing the final version of their manuscript:

1) page 1, line 29: "... without the occurence of redox." Redox what?

2) page 2, line 52: in the formula [YbCp2(OPPh3O)2] there is one oxygen too much

3) the 1H NMR is quite impacted by the paramagnetism of the trivalent Ytterbium (u= 4.5uB) leading to broad signals, but the 31P has a nice sharp singlet. Any explanation?

4) page7, line 191: "should be applicable to all rare earth elements". do the authors have any evidence for other lanthanide metals.?

5) page 7, line 210: perhaps mention that C6F5H and 1,2,4,5-tetrafluorobenzene are the major products and only small amounts of the meta product are formed

6) according to the authors complex 2 is formed via reduction of C6F5H by the divalent Yb intermediate. However, C6F5H is present only in small amounts in the reaction mixture, so oxidation to the trivalent complex 1 is the major pathway, leading to very small amounts of 2. Have the authors tried to add an excess of C6F5H right at the start to the reaction shown in scheme 5 to increase the amount of complex 2 by outbalancing the reduction of the ligand?

7) page 10, line 322: are the crystals of complex 1 polychromatic or colourless as mentioned in the text?

Reviewer 3 Report

The manuscript reports new syntheses to a homoleptic bis(diphenylphosphano)methanide lanthanoid(III) complex, including a redox step, and a case of facile C-F activation to a methanide-stabilised Yb-F cage.

The “protonated” neutral ligand bis(diphenylphosphano)methane dioxide is quite common in lanthanoid chemistry, whereas deprotonated “methanide” species are rare. Structures of these were surprisingly only very recently reported (ref 14) and this work described new synthetic “entries” into this substance class. It is a nice addition that the redox transmetallation approach works well here.

The coordination chemistry is well described in the paper, but Ln complexes with closely related bis(diphenylphosphano)methane disulfide/diimide ligands, often in their diide form, could be briefly introduced. It is worth noting that none of these structures feature three ligands per Ln centre as in the work here.

The work is well presented and described, and the supporting information provide very good background information with various spectra, including from in-situ and variable temperature experiments. I support publication of this work in Molecules after various minor changes, please see below, were considered.

The abstract states that complex 1 is “extremely” air-sensitive – is there a specific reason for this? It is a well-protected Yb(III) complex. This should be discussed.

What is the likely reason for the facile redox reaction for the formation of 1? Is the ligand so CH-acidic (is a pKa value known?) or is it linked to the “hardness” of the ligand?

The value in “…d8 -toluene, revealing a resonance at 4.79 ppm attributable to H2 gas formation”, and see the SI, should be commented on (and this can be done in the SI). One would expect a value closer to 4.50 ppm unless it is affected by paramagnetic species in the reaction mixture or another effect.

Regarding the calculated natural charges of complex 1 from computational studies, the value of the C(H) charge should be given in the main text. (Also, the value for O in Table S2 is likely negative which needs correcting). The calculated charge of the Yb centre is only +1.46 – is this expected and found for related complexes as well? It may be quite low for an Yb(III) compound and the applicability of the method should be checked here. The values should ideally also be briefly compared to the data reported in reference 14. It should be made clearer in the main text that the respective chemical shift of the 1H NMR resonance of the CH fragment is, however, likely not due to the calculated charge, but mainly due to the paramagnetism of the complex.

When the deviation from planes (c.f. Figure 4) and torsion angles (Table 1) are presented and discussed, approximate esd’s should be suggested to help the reader judge the significance of these differences. Interconversion between the isomeric forms may likely only require very low energy differences (below the limits of the T experiment); other geometric effects may be responsible for the splitting of resonances in the NMR spectrum and I agree with the authors that likely the various phenyl position are responsible. Where the conformational isomers observed in VT experiments are compared between the different Ln centres (and reference 14), the discussion should mention the ion sizes as a reason, if appropriate.

The reported X-ray diffraction equipment differs between the main text and the supporting information.

The SI details on the computational studies are quite short and should at least include atomic coordinates of the optimised structure.

Minor changes/typos:

Pentaflurobenzene etc … change to fluoro!

In 31P{1H}, the 1 needs to be superscript

The 1H NMR spectrum shows, not showed, plus related examples

Reviewer 4 Report

This excellent manuscript from Blair and coworkers reports the synthesis and characterization of two Yb(III) complexes with a bis(diphenylphosphino-oxide)methanide ligand. This kind of ligands are rare, shows a variety of different applications but are very scarcely studied and represented. The bibliography is adequated and the context of the research is updated. The synthesis is more or less clear, well explained (see comments) and the characterization seems to be correct. As a whole, the information contained in this manuscript is of interest for the readers of this journal and I recommend acceptation after some minor revisions.

- The reactions shown in Schemes 2 and 3 have to reflect the true stoichiometry of the reaction (at least some approximation). The authors have to adjust the stoichiometry. For instance, in Scheme 2 three ligands are deprotonated, but only two anionic Cp fragments are in the starting material, so how is it produced the third deprotonation? How many equivalents of CpH are produced? It seems that H2 is produced: is this related with the oxidation of the Yb from Yb(II) to Yb(III). Which is the oxidant? And the same for Scheme 3 (probably in the reaction shown in Scheme 4 this is not possible?)

- The color of complex 1 is curious: in method 1 (lines 318-330) it is obtained as polychromatic crystals, while in method 2 (lines 332-337) is colorless.

- Figure 2 should display the full spectra. Should it be possible to measure the 13C NMR spectra? This is just a question. It could be very informative to compare the 13C data with those reported in reference 14, because the 1H NMR data are quite different.

- There are minor typos that should be corrected: OPPh3O (line 52); 65% crystalline yield (??) should be isolation of crystalline 1 in 65% yield (line 64);
